# Iterative design of training data to control intricate enzymatic reaction networks

Bob van Sluijs[1], Tao Zhou ⬡[1] ✉, Britta Helwig[1], Mathieu G. Baltussen ⬡[1], Frank H. T. Nelissen[1], Hans A. Heus[1] & Wilhelm T. S. Huck ⬡[1] ✉

Kinetic modeling of in vitro enzymatic reaction networks is vital to understand and control the complex behaviors emerging from the nonlinear interactions inside. However, modeling is severely hampered by the lack of training data. Here, we introduce a methodology that combines an active learning-like approach and flow chemistry to efficiently create optimized datasets for a highly interconnected enzymatic reactions network with multiple sub-pathways. The optimal experimental design (OED) algorithm designs a sequence of out-of-equilibrium perturbations to maximize the information about the reaction kinetics, yielding a descriptive model that allows control of the output of the network towards any cost function. We experimentally validate the model by forcing the network to produce different product ratios while maintaining a minimum level of overall conversion efficiency. Our workflow scales with the complexity of the system and enables the optimization of previously unobtainable network outputs.

Living cells rely on enzymatic reaction networks (ERNs) to produce energy and building blocks to support cellular processes. Evolution has shaped these ERNs into interconnected sub-pathways to generate multiple outputs from multiple inputs, driving product formation across complex kinetic landscapes. Recently, significant progress has been made in reconstituting ERNs in vitro with the aim of building a cell from the bottom up[1–4], or to produce value-added chemicals from sustainable substrates as an advanced biotechnology[5–8]. However, most of these networks typically do not capture one of the essential features of biological ERNs, where several interconnected sub-pathways function simultaneously to generate multiple outputs. Controlling such networks remains challenging due to the lack of sufficiently informative experimental datasets that can be utilized to train kinetic models which trace the dynamic properties of large ERNs and enable on-demand design[9,10].

Typically, the optimization of ERNs towards specific outcomes, like increasing the overall efficiency, is achievable by searching a large combinatorial space of inputs and measuring the product formation of the ERN. Experimentally, this is prohibitively time-, labor-, and cost-intensive[11]. Recently Pandi et al. have shown that such a screening process could be significantly improved by an AI based active learning protocol[12]. Additionally, promising advances have been published

recently, utilizing machine learning to derive and individual reaction mechanisms from large datasets[9,10,13–15]. Yet, these black box approaches are limited in their ability to guide the design of large ERNs, however they are often very adept at mapping a specific region of the input output space, but not the entire space (the entire kinetic landscape). Kinetic models based on ordinary differential equations can track all the intermediates through time by explicitly formulated reaction rates and hence are especially powerful in guiding the optimization of complex ERNs[16]. In the context of larger networks, parameterizing these models is challenging. Not every interaction can be observed which complicates the identification of individual rates. Training data often relies on steady state batch experiments where a single combination of control inputs is tested. These experiments tend to be kinetically non-informative and are not sufficient to approximate the kinetic landscape of complex ERNs. To address this, time-course datasets which track the responses of ERNs to controlled perturbations are needed. This is demonstrated by both Shen et al. and Hold et al. in both batch and flow respectively, who characterized networks by adding the enzymes sequentially and measuring the change in product formation[17–19]. However, as the complexity and scale of an ERN increases (substrate competition, allosteric interactions, feedback loops, futile cycles, etc.) choosing a set of perturbations intuitively

[1]Institute for Molecules and Materials, Radboud University, Nijmegen, AJ, The Netherlands. ✉e-mail: tao.zhou@ru.nl; w.huck@science.ru.nl

such that we obtain relevant information about the kinetic landscape becomes increasingly difficult.

Here, we present a generalizable method that trains a kinetic model iteratively, by adding new and more informative experiments to a training dataset in each optimization cycle (akin to active learning). It incorporates an optimal experimental design (OED) algorithm that evolves a sequence of out-of-equilibrium perturbations to be maximally informative. We subsequently test the utility of the model by using the experimental outcomes of these perturbation experiments as test data for the previous iteration of the model. Using this approach, we demonstrate that a limited number of design iterations is enough to obtain data of sufficient quality to map the kinetic landscape of the ERN and obtain a measure of control over it as a multi-input multi-output (MIMO) system in vitro.

## Results

### Overview of the nucleotide salvage pathway

The in vitro ERN constructed in this work derives from the nucleotide salvage pathway (Fig. 1a), which regenerates nucleotides for cellular processes by recovering bases and nucleosides from the degradation of RNA and DNA. The network starts with phosphoribosyl pyrophosphate (*PRPP*), which can be converted from glucose via the pentose phosphate pathway and is coupled by the enzyme *UPRT* and *APRT* to nucleobases *uracil* and *adenine*, respectively, to form the monophosphate nucleotides *UMP* and *AMP*. For solubility reasons we did not include *guanine* as a nucleobase and started from *GMP*. *UMP*, *GMP* and *AMP* are subsequently converted to their corresponding diphosphate nucleotides (NDPs) by enzymes *UMPK*, *GMPK* and *AK*, respectively, using *ATP* as cofactor. Finally, NDPs are converted to NTPs by a single enzyme, *PK*. In total, this system consists of six enzymes catalyzing eight reversible reactions, where *PK* is shared between three substrates, and resource competition for *ATP*, *PEP* and *PRPP* throughout the network. Previous works demonstrated all these enzymes could function in one pot to synthesize labeled nucleotides with an excess

amount of the key compound[20–22], yet the overall performance is poor, controlling multiple state outputs remains a challenge, this requires the guidance of a kinetic model with sufficient resolution.

### Kinetic model of the nucleotide salvage pathway

Translating the reactions of the ERN into a coarse grained model of ordinary differential equation (ODEs), resulted in an ODE system of 15 equations with over 40 kinetic rates (for a full description of the model and coarse graining process see supplementary information 2). Generally, choosing the right model can be challenging (Fig S5-11), large enzymatic reaction networks require more parameterization, this can cause the model to overfit the training data, reducing its predictive power. This parameter problem is present in all models, but with ODEs it can be viewed from the perspective of a parameter's forwards sensitivity to the observed species (Fig. 1b)[23]. These sensitivities map onto the contribution a parameter has to the observed rates of change over time (Supplementary information 1.6eq.6). When these sensitivities correlate with one another, the observations can be approximated by the model by modifying both rates simultaneously. A positive correlation between the forward sensitivities of kinetic rates implies a similar effect on the rate of change of the observed species, thus the model can fit the data by increasing the value of one rate whilst decreasing the value of its partner, a negative correlation implies an opposing effect on the rate of change, thus, to fit the data the kinetic rates need both to either increase or decrease.

This unidentifiability means many combinations of kinetic rates can approximate the data (not just the 'true' rates), which in turn leads to prediction errors as the experimental conditions change from those used to generate the initial training data[23–25]. Thus, experimental data can be deemed uninformative if the inability to discern which reactions contribute most to the flux of a species at a specific time and results in prediction errors as

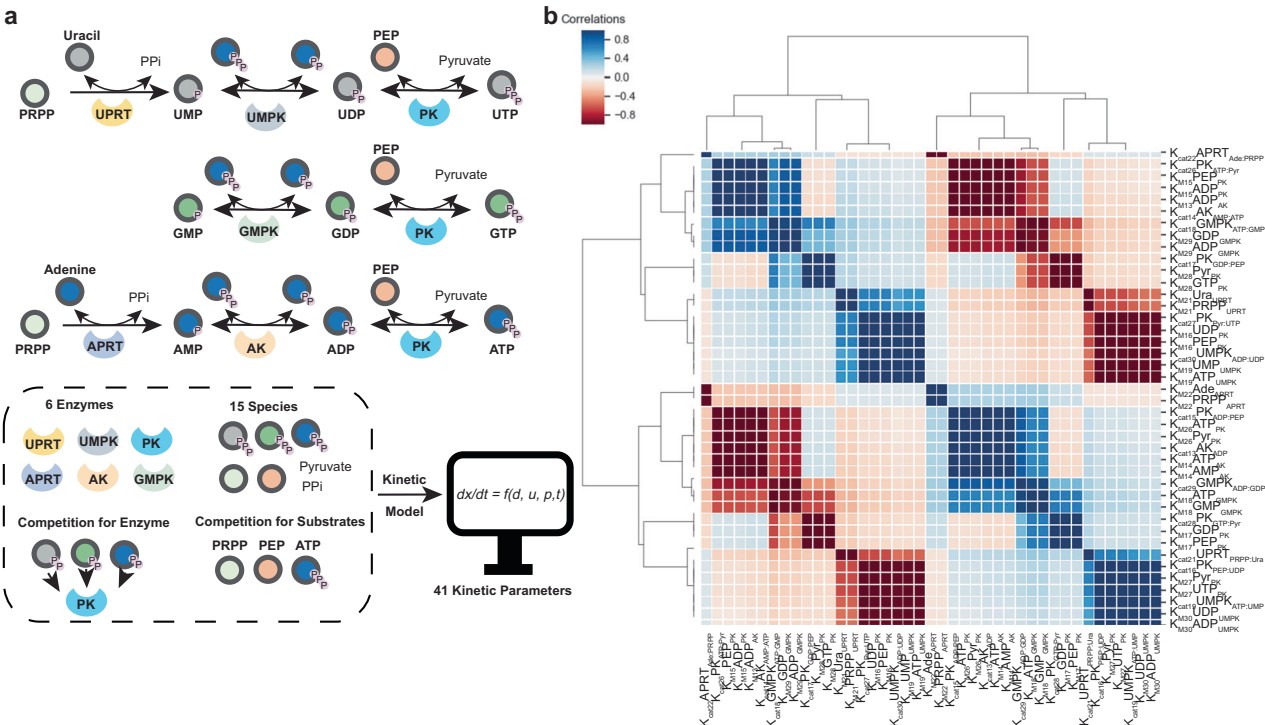

**Fig. 1 | Overview of the nucleotide salvage pathway and kinetic model parameter sensitivities to observed species. a** Reaction scheme of the in vitro reconstructed part of the nucleotide salvage pathway. The network consists of 6 enzymes and 15 substrates/products, resulting in a set of ODEs containing over 40 kinetic parameters. **b** Positive and negative correlations between the forward sensitivities between the kinetic parameters with respect to the measured output.

conditions change. Generally, it is easier to completely identify rates in simplified models, but their quantitative predictive power will be limited as mechanistic assumptions are readily broken (Supplementary information 2, Fig. S10). Conversely, detailed mechanistic models are more descriptive but it is harder to identify kinetic rates.

However, from a broadly practical perspective, precisely identifying individual rates is not needed to control the behavior of an ERN, a model just needs to approximate the kinetic landscape adequately and the remaining uncertainty needs to be manageable. To address this efficiently, we adapted an active learning approach commonly applied in machine learning with the singular goal of controlling ERNs. We utilized optimal experimental design (OED) to design experiments that maximize information about the ERN in the data, and subsequently train a kinetic model and tested its predictive power. This cycle was repeated until the uncertainty around the predictions was reduced and they matched the experimental outcome.

## OED and pulsing substrates into the flow reactor

We highlight this experimental workflow in Fig. 2a. First, all enzymes were individually immobilized on microfluidically produced hydrogel beads with a diameter of 50 μm[26]. The activity of each enzyme after immobilization was measured separately. Next, enzyme-loaded beads were loaded into a microfluidic continuous stirred-tank reactor (CSTR). The CSTR chamber itself has a volume of 100 μl and the flow setup has six inlets for each of the input substrates *uracil, GMP, adenine, ATP, PEP,* and *PRPP* and a single outlet (Fig. S15). Samples were collected from the output at different intervals depending on the total flow rates by a fraction collector and analyzed offline by ion-pair HPLC[27]. The analysis of the chromatographic peaks provides a compositional pattern of eight input substrate, intermediates, and product molecules (*uracil, UMP, GMP, adenine, ADP, GTP, UTP,* and *ATP*), each changing at every input combination.

The optimal experimental design workflow is shown in Fig. 2b. In step one a swarm/evolutionary algorithm evolves an input flow profile

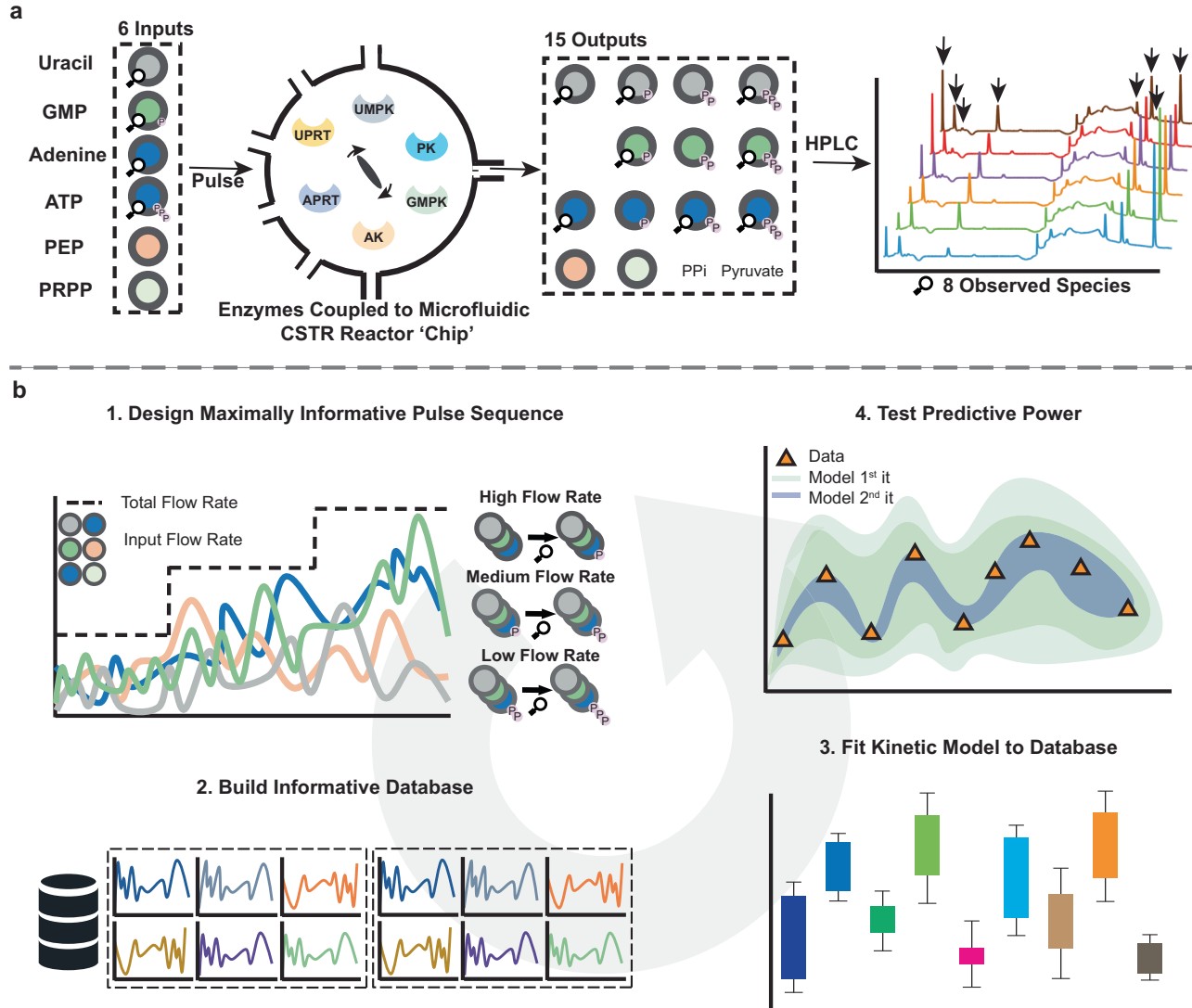

**Fig. 2 | Overview of the experimental flow set-up and the iterative design of training data to train a kinetic model. a** Schematic of the experimental workflow. Enzymes are immobilized on gel beads and placed in CSTR with 6 inlets containing different substrates The output is measured offline on an Ion paired HPLC, 8 species (*N* = 1, indicated by the arrows, from left to right: *uracil, UMP, GMP, adenine, ADP, GTP, UTP,* and *ATP*) can be observed over time. **b** Computational workflow to design an information dense dataset and train a kinetic model. In step one the OED

algorithm evolves control inputs (i.e. inflow rates of the 6 inlets) to be maximally informative. In step two this data is added to a training dataset which is subsequently used to fit a model in step three, resulting in a range of possible parameter values for each parameter (color). In step four we use the previous iteration of the model to predict the outcome of the latest experiment, utilizing this round as test data.

for each of the six inputs at three different flowrates[28,29]. This algorithm scores input patterns by maximizing the D-Fisher information criterion (Supplementary information 1.6eq.7)[30]. This criterion is obtained by computing the determinant of the Fisher information matrix which is derived from the parameter sensitivities (Supplementary information 1.2eq.6). This metric maps onto the volume of the parameter space where the ODE model can fit to the experimental data[30,31]. This means the algorithm is driven to find a combination of input sequences that breaks the correlation between parameter sensitivities (if only temporary). The transition between different total flow rates results in different output compositions and serves as another control parameter that increases the information content about substrate conversion fluxes in the data. At high flow rates input molecules and monophosphates are detected (as only a fraction of substrate has been converted); at low flow rates increased NTP formation is observed (Supplementary information 2.3 Fig. S13 & S14). In step two this data is added to a training dataset, the model is trained on this data in step three. In step four the predictive power of the model is assessed by using the previous iteration of the model to predict the current experiment (test data), if the predictive power is not sufficient or no longer improves, the cycle is terminated; if not, the cycle continues, and the latest iteration of the model and database is used to design a new experiment in step one[32].

### Iterative design of training data to build a kinetic model

A total of three iterations of the optimization cycle were performed (excluding a calibration), each time exchanging the microfluidic chip, altering the enzyme concentrations (Supplementary information 3.2 Fig. S17-S20). The lower and upper boundary of the concentration ranges for the substrates were based on the enzyme activity assays and substrate solubility (Supplementary information 4.4 Fig. S24-S37). The initial experiment (not part of the cycle) is manually designed and 'calibrates' the model (Supplementary information 3.2 Fig. S17). This allows for the subsequent OED of an informative input sequence since more knowledge about the system equates to better OED outcomes[26]. To illustrate the non-intuitive character of the evolved input sequence we show the substrate inputs of the final experiment of the optimization cycle (Fig. 3a) and the complexity of the time-course data including model convergence (Fig. 3b).

We subsequently place these data in the context of the optimization cycle (Fig. 4). Figure 4a shows parameter distributions of the model trained in the first iteration (top) and the parameter distribution of the model trained in the third iteration (bottom). We note a significant decline in the distribution width of most kinetic parameters (Fig. S14). To demonstrate the improved predictive power of the model, Fig. 4b compares the predicted outcomes (shaded area) of the model trained after iteration one and iteration two of the OED cycle (predicting the experiment performed in the third iteration shown in Fig. 3). The second iteration of the model already shows a drastic reduction in the variance around the prediction and highlights that the model can approximate the behavior of the ERN quantitatively.

### Trained model controls nucleotide salvage pathway in flow

This presents us with new opportunities for the third iteration of the model, beyond traditional optimization schemes that often focus on maximizing the yield of a single product. Here, we demonstrate how we can use the final iteration of the model to control a MIMO system to achieve a range of more complex output states[29,33]. We opted to tune the *ATP/GTP/UMP* output ratios whilst maintaining a minimal conversion efficiency−defined as the percentage of nucleobases converted to triphosphates−of 60%.

The outcome of this sampling process is shown in Fig. 5a, we randomly generated $10^5$ substrate input combinations, each input combination was simulated twenty times using different combinations of estimated kinetic rates. Every dot represents a different condition,

the color indicates the ratio between *ATP/UTP/GTP*. The 20 sets of estimated kinetic rates –when simulated- predict different *ATP, UTP,* and *GTP* concentrations. This is reflected by the y-axis which shows the standard deviation of the predicted mean concentrations for these simulations. It captures the certainty of the model and the likelihood there will be a prediction error for a given set of input conditions. The x-axis subsequently shows the conversion efficiency. We selected seven experimental conditions representing seven *ATP/UTP/GTP* ratios in Fig. 5a, including one repeated ratio (experiment 1 & 7) and one experiment with a lower conversion efficiency (experiment 3). This experiment serves two purposes: first, to demonstrate that the model can control a MIMO system and access a part of the output space that requires an accurate map of the kinetics and finely tuned control inputs (which is achieved by optimizing the ERN for different triphosphate blends with a high conversion efficiency). Second, to identify the operable space of the model, for which we test a range of total input substrate concentrations along with compositional blends of final products.

Figure 5b shows the predicted confidence interval of the final yield, and the yield as measured on the HPLC. For experiments 1-5 uncertainties and total output concentrations vary but predictions still match. For very low input concentrations of *UMP, guanine, adenine,* and *ATP* in experiments 6 and 7, the predictions error increases even though the simulated standard deviation is low. This relation between the prediction error, quantified as the percentage the simulated mean deviates from the HPLC measurement and the summed input concentration of the nucleobases is shown in Fig. 5c. It highlights that the model can predict exact concentrations as long as the total concentration of substrate inputs is larger than 0.3 mM. The cause can likely be attributed to a decrease in the signal to noise ratio for the HPLC measurement, leading to larger variations in the experimental data (see Supplementary information table S1-2). To test this, we used different models. More complex models which contained different rate laws (Fig. S5-6) as well as combinations of allosteric interactions reported in literature (Fig. S7). However, none of these models performed better and prediction errors increased. This suggests that these interactions do not play a significant role in this network, at least not significant enough to overcome a potential overfit of the training data. In contrast, reducing the complexity of the model increased the prediction error significantly, we were able to confirm that reactions catalyzed by the *PK, UMPK, AMPK,* and *AK* enzymes need to be reversible, whereas *UPRT* and *APRT* can be considered unidirectional (Fig. S8-11). In summary, this means that a total input concentration 0.3 mM marks the practical boundary of the model trained on this data, knowledge we can leverage to efficiently probe conditions in the identified operable space.

## Discussion

We have presented a methodology to design informative training data and map the kinetic landscape of an ERN as efficiently as possible. By designing sufficiently complex experiments we were able to restrict the combinations of potential kinetic rates such that they map onto real product formation fluxes across a large input-output space. This space could subsequently be sampled for any cost function. To highlight this versatility, we opted to create different compositional blends of triphosphate compounds which require not one but multiple finely tuned input conditions. Finally, we identify the operable space wherein the model is useful and demonstrate that other mechanistic descriptions of the systems reduce the predictive power of the model. This underscores that the active learning aspect of the OED pipeline is able to balance the degree to which we parameterize the model, its mechanistic assumptions, and its predictive power within three iterations.

The number of OED iterations required to achieve this depends on both the complexity of the network and the quality of the experimental data. If the system is highly non-linear, more certainty about the rates

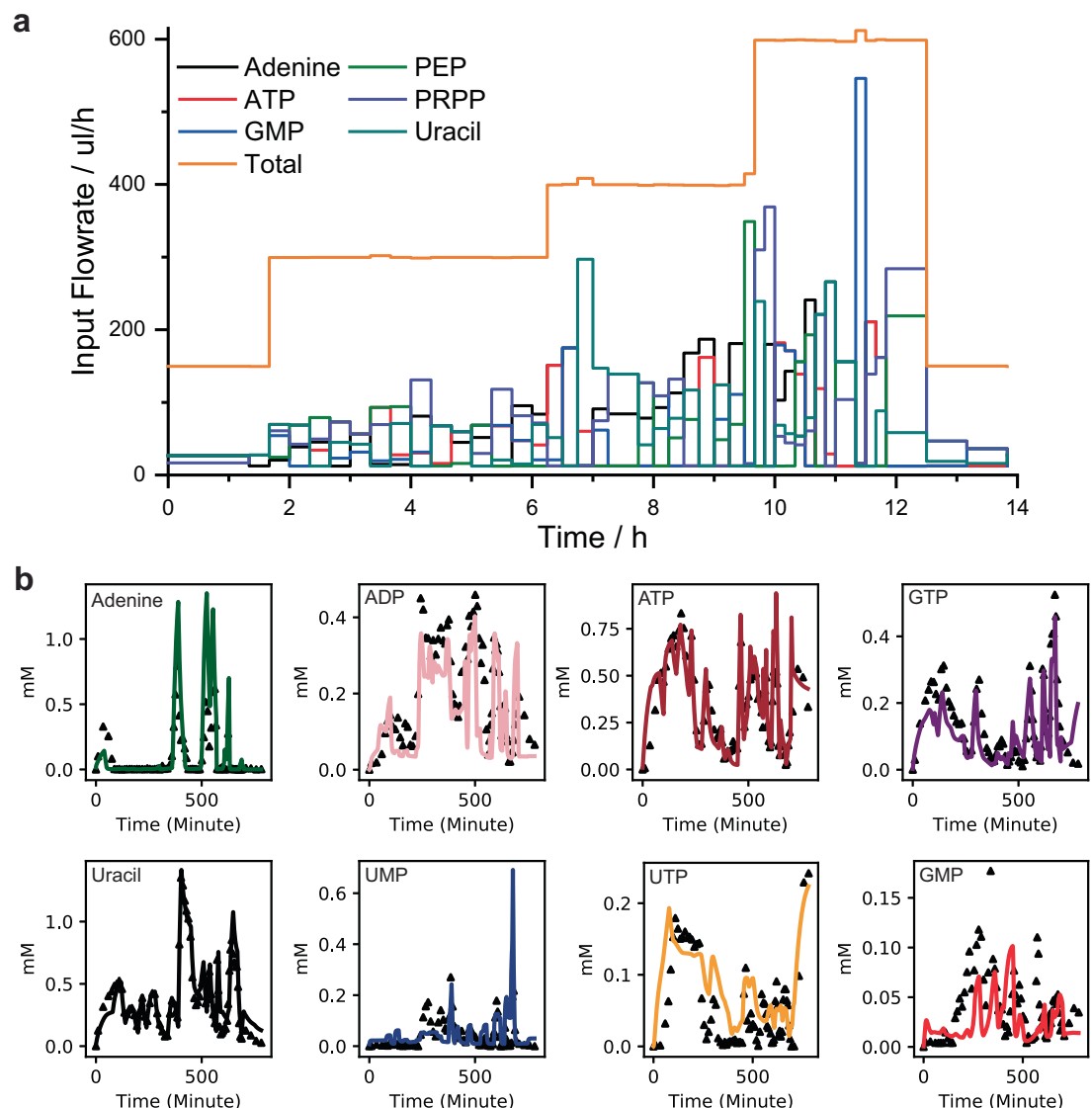

**Fig. 3 | Example of optimally designed flow profile and measured output (3ʳᵈ iteration). a** Input flow rates for each of the 6 inputs substrates evolved by the OED algorithm. **b** Data as measured on HPLC (black triangles) and the fit of the model to the data (solid lines), Source data are provided in Source Data Fig3.

will be needed as smaller deviations from the true value will result in larger prediction errors. In contrast, very linear and orthogonal networks will likely require significantly fewer optimization cycles (and a simpler model) to enable a form of MIMO control. Overall, this means the pipeline can be utilized in different contexts as long as there is a kinetic model with control inputs (in Fig. S1 we probed the applicability of this software to larger systems, specifically by comparing CPU time needed for the model presented here and the *E. coli* core metabolism model). Process optimization for organic synthesis using design of experiments in flow has been reported, most of which aim to determine the optimal operational conditions for one reaction[34–36]. So far experimental design schemes have not been applied to multiple organic reaction networks (or placed in the context of an active learning cycle). However, there is no reason why it cannot be applied to train a kinetic model which provides more understanding and a high-level of control over chemical reaction networks[10,13].

In future work, more complex cost functions can be defined, including the identification of key reaction mechanisms and interactions by making the coarse graining process of the model an explicit part of the active learning process. In this instance, the algorithm—besides mapping the kinetic landscape- seeks to find input

combinations which either validate or invalidate mechanistic assumptions embedded in different models. Currently, we are able to discriminate between different rate laws (broadly classifying them as descriptive or not) and the inclusion of reaction reversibility, whereas potential allosteric interactions did not seem to be present in a manner that effected predicted outcomes. However, the differences were not explicitly maximized by the algorithm, thus the observed difference in predictive power was minimal in most cases[37]. Nevertheless our results are promising, and are complimentary to other work that has shown that black box models can identify the reaction mechanism of a single reaction from bulk data[14]. Such approaches have not been reported in the context of a biochemical network nor have they been embedded in an active learning like approach which offers promise for the future. Overall, we believe our pipeline is beneficial to all who seek to build complex biochemical pathways with controlled inputs.

## Methods
### Materials
Enzymes adenylate kinase (*AK*) and pyruvate kinase (*PK*) and all chemicals were purchased from Sigma and directly used without further processes. Enzymes adenine phosphoribosyl transferase (*APRT*) and,

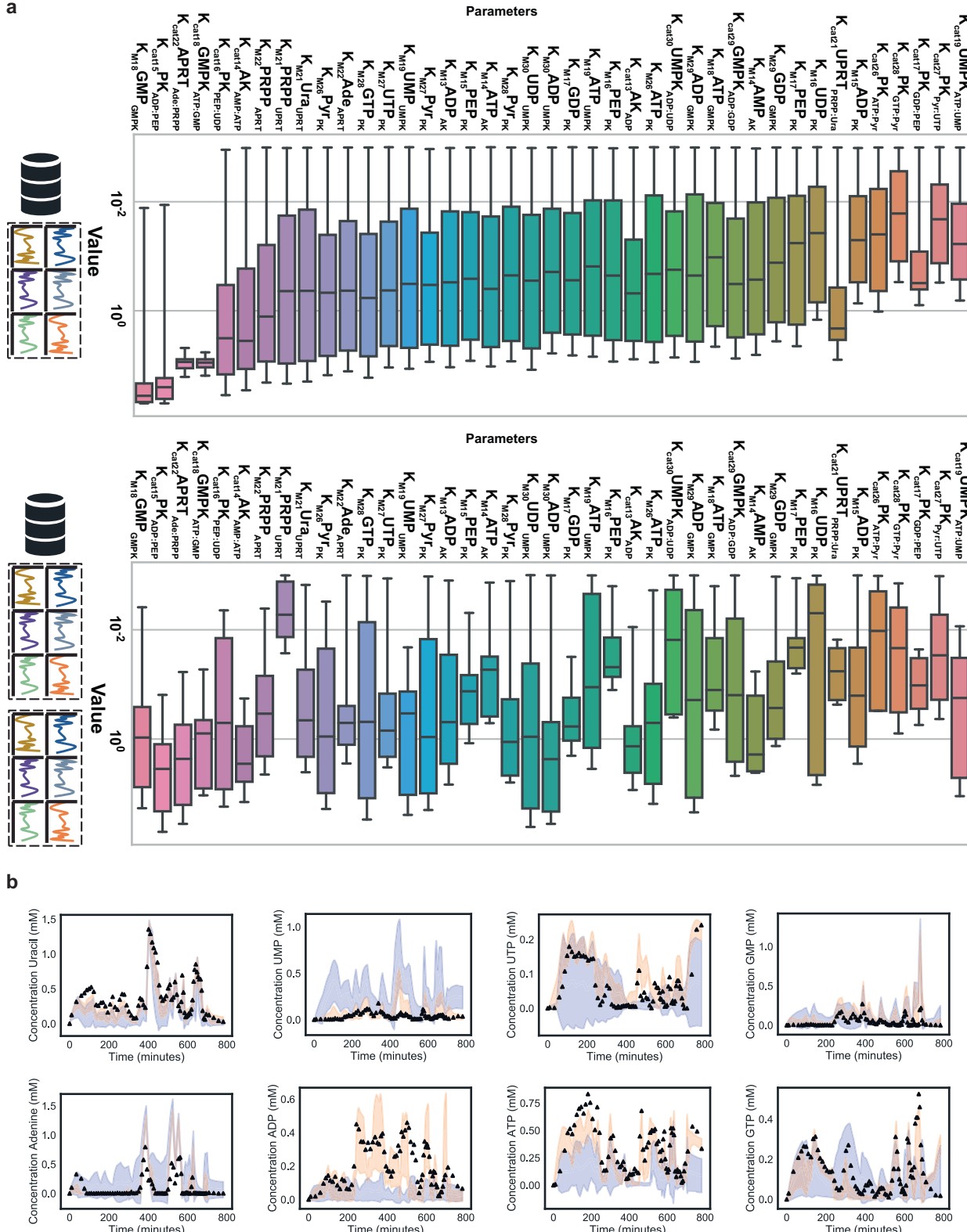

**Fig. 4 | Application of the iterative design of training data and its impact on identifiability and the predictive power of the model. a** Distribution of fits of the parameters including either only the first (number of datapoints $N = 211$) or the 3rd iteration (number of datapoints $N = 166$, the box itself shows the quartiles where middle boxes represent 50%, with a line showing the median value. The whiskers of the box show the highest and lowest values), the parameter set is included if the fit score deviates no less than 15% from the best fit, the y-axis denotes the parameter value, the catalysis rates are in mM/min, the Km values in mM. We note that after new rounds are added the distributions of the parameters decreases. **b)** Prediction of the last experiment (black triangles) in the dataset using the model trained on the dataset obtained after the first (shaded blue) and second (shaded orange) iteration of the cycle, we simulated the model using the best parameter sets ($N = 20$), the shade area reflects the standard deviation around the mean prediction. Source data are provided in Source Data.

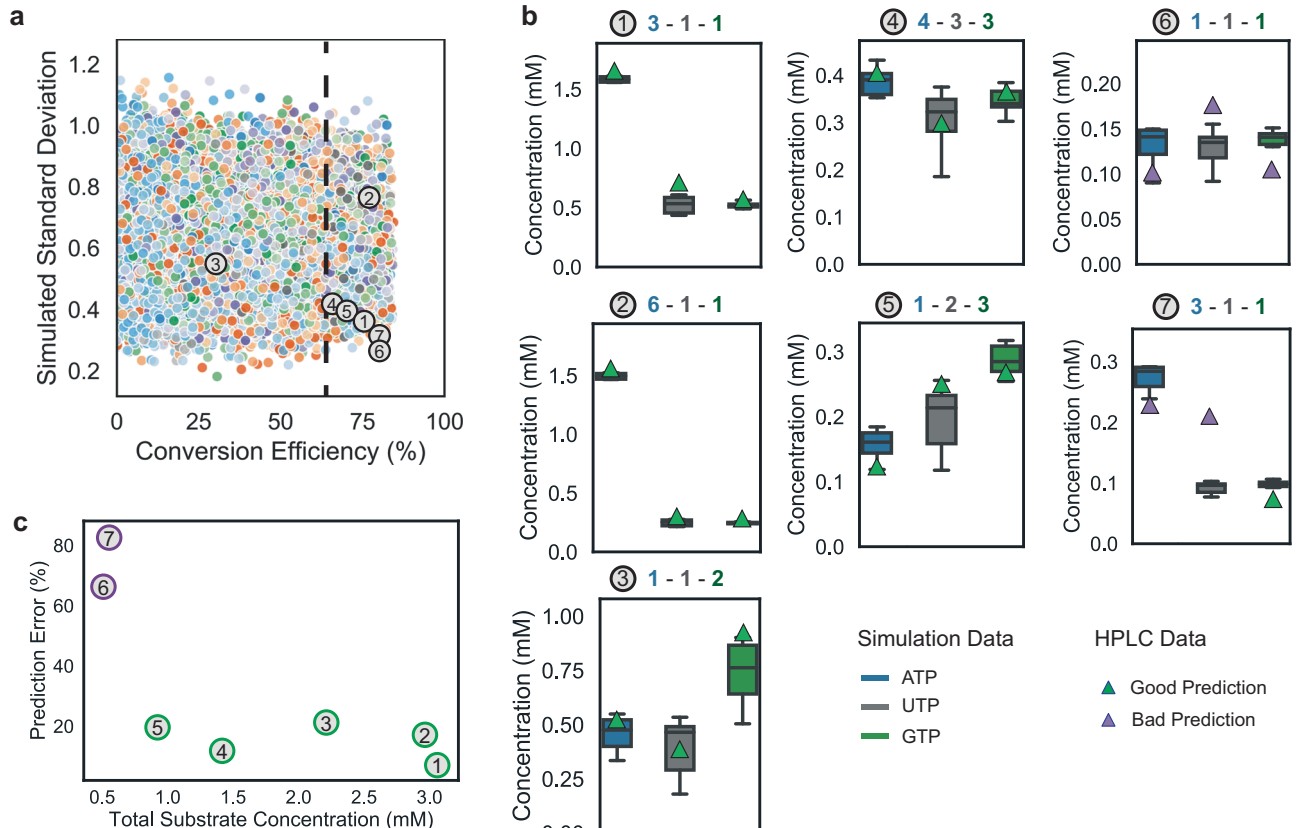

**Fig. 5 | Controlling the nucleotide salvage pathway as a MIMO system and testing the model by predicting product ratios. a** Shows the range of possible different ratios given different substrate inflow rates, we opted to screen a large space of experimental and select 7 ratios (number in circle) to test along a range of summed substrate input concentrations (numbered spots). Each color is a simulated ratio, the standard deviation is the simulated deviation around the predicted mean (y-axis). The conversion efficiency is the predicted fraction of nucleobases that is converted to a triphosphate. To calculate the efficiency of the *adenine* conversion we first subtract the *ATP* concentration input from the measured *ATP* output. **b)** shows the experiments, labeled 1-7, with both the simulated concentrations including confidence interval (N = 20, The box itself shows the quartiles where middle boxes represent 50%, with a line showing the median value. The whiskers of the box show the highest and lowest values) and the HPLC measurement. The ratio between *ATP* (blue), *UTP* (gray), and *GTP* (green) are shown on top. **c)** Shows the prediction error defined as the percentage the simulated mean deviates from the HPLC data on the y-axis (averaging the error for the three triphosphates) and the total concentrations of the input substrates on the x-axis. Source data are provided in Source Data.

uracil phosphoribosyl transferase (*UPRT*) were expressed and purified as described by Arthur et al.[38], Genes for guanosine monophosphate kinase (*GMPK*) and, uridine monophosphate kinase (*UMPK*) were PCR amplified from *E. coli* K12 using gene specific primers, cloned into pET15b, expressed overnight at 30 °C (*GMPK*) and 18 °C (*UMPK*) in *E. coli* BL21(DE3) and purified according to protocols modified from Oeschger et al. [39] (*GMPK*) and Serina et al. [40] (UMPK) to accommodate Ni[2+]sepharose purification. Purified enzymes were dialyzed against 20 mM potassium phosphate buffer (pH 7.2) prior to immobilization. All the enzymes were immobilised on microfluidic produced hydrogel beads, as reported[26]. After immobilization, all the enzyme-beads were freeze dried and stored in -20 °C. 1 mg of beads for each enzyme was suspended in 31 ul IVTT buffer (pH 7.3, 9 mM magnesium acetate, 5 mM potassium phosphate, 95 mM potassium glutamate, 5 mM ammonium chloride, 0.5 mM calcium chloride, 1 mM spermidine, 8 mM putrescine, 1 mM dithiothreitol, 10 mM creatine phosphate). All reactions were conducted in this so-called IVTT buffer at room temperature.

### Flow experiments setup
Cetoni Nemesys syringe pumps with Hamilton syringes were used to control input and the flow profile was programmed using the Cetoni neMESYS software[26,41]. Before performing the designed flow profile, the whole system was equilibrium with buffer for two hours. The outflow of the CSTR was collected using a fraction collector, collecting

for either 30 or 15 minutes or three droplets per fraction. The ion-pair HPLC analysis was adapted from ref. 26 and performed on Shimadzu Nexera X3 HPLC system with an Inertsil ODS-4 column (3 μm, 150 × 4.6 mm; GL Science) and a guard column (3 μm; 10 × 4.6 mm) at 40 °C. The elution gradient was as follows: 100% buffer A (100 mM potassium phosphate buffer (pH 6.4) with 8 mM ion-pair reagent tetrabutylammonium bisulfate, filtered before use) for 13 min; 0–77% linear gradient of buffer B for 22 min; 77–100% buffer B (70% buffer A with 30% acetonitrile) for 1 min; and 100% buffer B for 14 min. The flow rate was maintained at 1 ml/min. Peaks were identified by comparison with standard samples. The concentration was obtained from the integrated peak areas with the calibration curve of each standard.

### Software and modeling
An overview of the software that performs the optimizations can be found in Supplementary Information 1. A generated text-based model object[25] is translated to an SBML and AMICI object modified from ref. 28 and ref. 42 (Supporting information Fig. S1-S4). AMICI is an ODE compilation package to C + + which is continuously updated[43–46]. Several publicly available tools integrate with AMICI[45–49]. This is needed for the expanding repertoire of ever larger kinetic models (most in vivo)[50–54]. To quantify the computational cost (and its general application to larger systems) we tested the speed of the pipeline presented on an in vivo metabolic core *E. coli* core metabolism model (ref. 53) and placed it in the context of our in vitro reactor set-up (see

Fig. S1). This test was run on a single core of Intel Xeon E5-1660 v4 @ 3.2 GHz. For more information on the efficiency of AMICI itself (where the bulk of the calculations are performed), we refer the reader to refs. 43,44,55–57, or its, by now, numerous applications[58–64].

## Statistics & reproducibility

No statistical method was used to predetermine the sample size. No data were excluded from the analyses; for the experimental data shown in Fig. 5, the experimental conditions predicting specific ratios were selected randomly after sampling $10^5$ possible ratios of *ATP/UTP/GTP*. Provided these ratios conformed to the required conversion efficiency (60%) and the chosen set of conditions differed sufficiently between the summed total inflow concentration of all substrates to cover the largest possible space and test the model.

## Reporting summary

Further information on research design is available in the Nature Portfolio Reporting Summary linked to this article.

## Data availability

All relevant data supporting the key findings of this study are available within the article and its Supplementary Information files. Source data are provided with this paper as a singular source data file, including the time-dependent inputs and HPLC quantifications and parameter estimates, archive https://doi.org/10.5281/zenodo.10411170. Source data are provided in this paper.

## Code availability

The package is written in Python 3.8 (python software foundation, Delaware US). Code can be found at Huckgroup GitHub at http://github.com/huckgroup/OED, code archived (see ref. 65), https://doi.org/10.5281/zenodo.10411170 (2023). For more information contact bob.vansluijs@gmail.com.

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

## Acknowledgements

This project is funded by the European Research Council (ERC) under the European Union's Horizon 2020 research and innovation programme (ERC Adv. Grant Life-Inspired, grant agreement No. 833466 and ERC PoC Grant OptiPlex, grant agreement No. 101069237). T. Z. acknowledges the Swiss National Science Foundation for financial support (P500PB_203166).

## Author contributions

B.v.S., T.Z. and W.T.S.H. conceived the study. B.v.S and T.Z designed and performed experiments respectively. B.v.s., T.Z. and W.T.S.H. analyzed the data and discussed the results. B.H. carried out foundational work and B.v.S. and M.G.B. built software to auto- generate strings for kinetic models of ERNs. F.N. and H.H. purified the four commercial unavailable enzymes provided the related plasmids. All authors discussed the results, provided comments, and revised the manuscript.

## Competing interests

The authors declare no competing interests.
