## [Peer Review File · Nature Communications]

Iterative Design of Training Data to Control Intricate Enzymatic Reaction NetworksEditorial Note: This manuscript has been previously reviewed at another journal that is not operating a transparent peer review scheme. This document only contains reviewer comments and rebuttal letters for versions considered at *Nature Communications*.

Reviewer #1 (Remarks to the Author):

The authors have addressed all my concerns. The quality of the revised manuscript has been significantly improved.

Reviewer #3 (Remarks to the Author):

This manuscript has been significantly revised and the authors have taken great care in clarifying the content to make it easier to read and follow. For what concerns my raised points, the authors have completely clarified the immobilisation strategy adopted and the stability of the immobilised enzyme system. I am satisfied that these aspects are now completely resolved.

Reviewer #4 (Remarks to the Author):

The paper describes an active learning like method to obtain sufficient data to fit a kinetic model for a multi-enzyme reaction in flow. This aspect of the work appears impressive and I believe would be of interest to a specialist audience. The authors go on to demonstrate that this model can be used to optimise reactions towards desired outcomes. The specific multi-enzyme reaction the authors have chosen appears to not be a major concern for this paper, as it is not discussed in either the introduction or the conclusions.

I still have concerns with this work around the enzymology and usefulness of the model. However, the novelty and technical accomplishment of the active-learning modelling approach are perhaps more important, warranting its publication in Nature Communications.

Some more specific comments follow:

The title seems better. However, my point about the use of the word design in the main text remains. Personally, I see the design of a reaction being the selection of which reactions/enzymes to use – with the modelling presented in this work being reaction optimisation. However perhaps this is a more subjective point, if the authors are set on describing this as design, so be it.

It would be a good test of the model to see if it can generalise to a new reactor or modes (like a batch reaction, or a scaled-up flow reaction), as suggested by reviewer 2. Presumably if the kinetic parameters (and model structure) identified are not just apparent parameters specific to the experimental context, this should be possible. I suppose the authors are either fitting a kinetic model which could be used reliably in a new context, with meaningful kinetic parameters (albeit for the beads rather than free enzyme). Or they're fitting a model with parameters which are only relevant in the context in which they have been determined.

Presumably these enzymes have been studied previously using traditional enzymology techniques, which could be used to suggest model structure, and perhaps any necessary additional terms for inhibition ect? For example, a quick search tells me that “.UPRTase of E. coli obeyed the kinetics of a sequential mechanism with the binding of PRPP preceding the binding of uracil.” <https://doi.org/10.1021/bi982279q>. In contrast, the authors get the worst fit modelling everything as ordered – but likely not all the enzymes have the same type of mechanism. Instead, could all rate law combinations be considered and scored for goodness of fit? Or if that's too computationally expensive, perhaps rate law selection could be considered as a parameter to be optimised during model fit? Indeed, this could validate the argument in the conclusions - “Currently we were already able to discriminate between the usage of different rate laws, the addition of allosteric interactions and the inclusion of reaction reversibility” if the ‘correct’ rate laws are identified?. Indeed, it doesn't seem that the model identified the allosteric activators identified by reviewer 2?

The authors haven't really answered my question about enzyme concentration in the reactor. Do the authors have some idea of a ballpark for figure for enzyme concentration – this doesn't need to be precise. A simple measurement for protein concentration (for example by absorbance at OD280nm) before and after immobilisation would provide as estimate for this. And then importantly, is this well below substrate concentration? If not, one of the key assumptions for Michaelis-Menten kinetics is invalidated and the authors should discuss this in the text.

The authors claim HPLC error is the reason for poor predictions below 0.3 mM. It might be helpful to show a standard curve and perhaps a couple of representative traces at 0.3 mM and below to back this up. Perhaps more importantly, if there is indeed poor signal to noise below 0.3 mM, this should be apparent from multiple technical replicates for experiments 6 and 7? Do the authors have such data? Otherwise, it does seem more likely to me that the model fails at these low concentrations because the model structure is not quite right (as highlighted by myself and reviewer 2), but that this isn't important at higher concentrations (ie when operating above K_m).

Quite a lot of the data used to fit the model in Figure 3b appears to be below 0.3 mM, is this not an issue if HPLC signal to noise is a problem?

Reviewer #5 (Remarks to the Author):

In this manuscript, the authors have proposed a methodology based on active learning to identify optimal datasets to train kinetic models of enzymatic reaction networks. They applied this method to a network containing 8 reactions to modulate product ratios while maintaining the overall conversion efficiency.

Major comments:

1. Novelty: The focus of the manuscript appears to be on optimal experimental design, i.e., identifying the most informative datasets that maximize predictive capability. It is unclear what this method does differently than a simple k-fold cross validation that is routinely used to validate kinetic models of metabolism. Furthermore, kinetic models of reaction networks typically contain far more parameters than data available for parametrization. The authors must also clarify how their OED algorithm ensures statistical significance of the identified parameters (passing a χ^2 goodness-of-fit test) to avoid overfitting the training data.
2. Applicability to larger networks and other kinetic formalisms: The authors have applied their algorithm to a small model containing 8 reactions and 15 metabolites formulated using only the Michaelis-Menten type formulation. Several other formalisms are commonly used in the realm of kinetic modeling (Saa and Nielsen, 2017) that have a varying number of parameters. Small models such as the one presented in this manuscript are typically well behaved and do not contain the complex interactions that contribute to parameter identifiability issues that can adversely impact predictive capability. The authors must evaluate the compatibility of their method with other modeling formalisms and quantify the influence of model formalism on identified optimal datasets. The authors must also apply their method on larger metabolic models such as those available for *E. coli* and other toy networks (Dash et al., 2017; Gopalakrishnan et al., 2020; Khodayari et al., 2014).
3. Computational scalability: The authors have not provided any details on the computational power and time required to identify the optimal training dataset. The authors must also evaluate the performance of their algorithm with larger networks. Examples of networks at various sizes are available from literature (Gopalakrishnan et al., 2020).
4. Uncertainty analysis and quantification: Parameter identifiability and precision is a common issue with kinetic models due to their nonlinear and nonconvex structure. Although the authors have focused on product yield as their validating data, poor parameterization precision can adversely affect the predictive power, especially in models with perturbed enzyme levels. Figure 4a

shows a poor resolution of K_{cat} and K_M parameters for most reactions. This will impact the predictive power of networks in which enzyme levels are perturbed, which was not done in this study. The authors should include additional validation for experiments with perturbed enzyme levels to reinforce their analyses.

References

- Dash, S., Khodayari, A., Zhou, J., Holwerda, E.K., Olson, D.G., Lynd, L.R., and Maranas, C.D. (2017). Development of a core *Clostridium thermocellum* kinetic metabolic model consistent with multiple genetic perturbations. *Biotechnol Biofuels* 10, 108.
- Gopalakrishnan, S., Dash, S., and Maranas, C. (2020). K-FIT: An accelerated kinetic parameterization algorithm using steady-state fluxomic data. *Metab Eng* 61, 197-205.
- Khodayari, A., Zomorodi, A.R., Liao, J.C., and Maranas, C.D. (2014). A kinetic model of *Escherichia coli* core metabolism satisfying multiple sets of mutant flux data. *Metab Eng* 25, 50-62.
- Saa, P.A., and Nielsen, L.K. (2017). Formulation, construction and analysis of kinetic models of metabolism: A review of modelling frameworks. *Biotechnol Adv* 35, 981-1003.

Reviewer #1 (Remarks to the Author):

The authors have addressed all my concerns. The quality of the revised manuscript has been significantly improved.

Reviewer #3 (Remarks to the Author):

This manuscript has been significantly revised and the authors have taken great care in clarifying the content to make it easier to read and follow. For what concerns my raised points, the authors have completely clarified the immobilisation strategy adopted and the stability of the immobilised enzyme system. I am satisfied that these aspects are now completely resolved.

Reviewer #4 (Remarks to the Author):

The paper describes an active learning like method to obtain sufficient data to fit a kinetic model for a multi-enzyme reaction in flow. This aspect of the work appears impressive and I believe would be of interest to a specialist audience. The authors go on to demonstrate that this model can be used to optimise reactions towards desired outcomes. The specific multi-enzyme reaction the authors have chosen appears to not be a major concern for this paper, as it is not discussed in either the introduction or the conclusions.

I still have concerns with this work around the enzymology and usefulness of the model. However, the novelty and technical accomplishment of the active-learning modelling approach are perhaps more important, warranting its publication in Nature Communications.

Some more specific comments follow:

The title seems better. However, my point about the use of the word design in the main text remains. Personally, I see the design of a reaction being the selection of which reactions/enzymes to use – with the modelling presented in this work being reaction optimisation. However perhaps this is a more subjective point, if the authors are set on describing this as design, so be it.

We understand that there can be ambiguity. We therefore changed the title to:

Iterative Design of Training Data to Control Intricate Enzymatic Reaction Networks

Changed in abstract

yielding a descriptive model that allowed design to yielding a descriptive model that allowed control

Our workflow scales with the complexity of the system and enables the design of previously unobtainable network outputs. to Our workflow scales with the complexity of the system and enables the optimization of previously unobtainable network outputs.

In the introduction

The design of such networks remains challenging to Controlling such networks remains challenging

and

Typically, the design of ERNs to Typically, the optimization of ERNs

It would be a good test of the model to see if it can generalise to a new reactor or modes (like a batch reaction, or a scaled-up flow reaction), as suggested by reviewer 2. Presumably if the kinetic parameters (and model structure) identified are not just apparent parameters specific to the experimental context, this should be possible. I suppose the authors are either fitting a kinetic model which could be used reliably in a new context, with meaningful kinetic parameters (albeit for the beads rather than free enzyme). Or they're fitting a model with parameters which are only relevant in the context in which they have been determined.

The model that is trained is indeed bead specific. The combination of the flow set-up with the beads eases downstream processing and uses fewer inputs. Though batch reactions were not explicitly tested, the model should be applicable to batch conditions since it is trained on low 'pseudo batch' flow rate ~1 reactor volume replaced per hour (slow compared to conversion rates) and very high flow rates (~6 reactor volumes). In the new model building software we have now included the capacity to model the enzymes in a different reactor setting, including multistate reactors.

Presumably these enzymes have been studied previously using traditional enzymology techniques, which could be used to suggest model structure, and perhaps any necessary additional terms for inhibition ect? For example, a quick search tells me that "...UPRTase of E. coli obeyed the kinetics of a sequential mechanism with the binding of PRPP preceding the binding of uracil." <https://doi.org/10.1021/bi982279q>. In contrast, the authors get the worst fit modelling everything as ordered – but likely not all the enzymes have the same type of mechanism. Instead, could all rate law combinations be considered and scored for goodness of fit? Or if that's too computationally expensive, perhaps rate law selection could be considered as a parameter to be optimised during model fit? Indeed, this could validate the argument in the conclusions - "Currently we were already able to discriminate between the usage of different rate laws, the addition of allosteric interactions and the inclusion of reaction reversibility" if the 'correct' rate laws are identified?. Indeed, it doesn't seem that the model identified the allosteric activators identified by reviewer 2?

We agree with the reviewer about the scope of possible models within, even within this network. If we included all possible combinations of rate laws for 5 enzymes, and all potential allosteric interactions, we would have a combinatorial explosion of testable models (over 1 million models), which requires a different approach. In the appendix we added an additional section on the computational time window the software operates in on a single CPU - see response to question 3 reviewer 5 or figure S1. Though AMICI is fast, testing all possible models is currently beyond the scope of this paper. Instead, here we first demonstrate that placing the OED algorithm in a design build test cycle, we can rapidly train our model up to the point where we have experimental control over the reaction.

In a follow up we can expand the algorithms utility by focusing on rate law identification and/or 'hidden' allosteric interaction specifically. Thus, we wholeheartedly agree with the reviewer, a next step would indeed be to include model identification within the algorithm. For that to work we likely – need to use free enzymes (costlier) and continuously change their concentrations during the experiment, and not between experiments like we do now.

The examples chosen, e.g. ping pong, ordered reaction, generalized Hill and rapid equilibrium random, highlight that at least differences for trained models do exist for this dataset, both when fitting and predicting state outcomes. Of these, ordered reaction models fit worse than the other three and were thus eliminated, and also ping pong models did not predict as well. Generalized hill equations are effectively a nested version of the rapid equilibrium random kinetic, but they do eliminate a parameter. Hence it makes sense that these models perform similarly. In contrast, removing a reaction i.e. reversibility, has major consequences for the predictive power of the model. If, for example, PK is considered irreversible, then this trained model is biased because it can no longer predict state outcomes. For the allosteric interactions, we can only say that these potential interactions were not sufficiently prominent to have an effect on the model predictions.

Alluding to potential future work, we can do a multi-objective optimization OED, that not only maximizes the information about the kinetics along an information criterion but also designs pulses which maximize the difference between the predictions of each proposed model across iterations (after which we discard options which neither fit and or predict). This however, is only useful if the trained models are quantitatively accurate in the first place.

We changed the sentence

“Currently we were already able to discriminate between the usage of different rate laws, the addition of allosteric interactions and the inclusion of reaction reversibility”

to

Currently we were already able to discriminate between the usage of different rate laws (broadly classifying them as descriptive or not) and the inclusion of reaction reversibility, whereas potential allosteric interactions did not seem to be present in a manner that effected model predictions.

The authors haven't really answered my question about enzyme concentration in the reactor. Do the authors have some idea of a ballpark for figure for enzyme concentration – this doesn't need to be precise. A simple measurement for protein concentration (for example by absorbance at OD280nm) before and after immobilization would provide as estimate for this. And then importantly, is this well below substrate concentration? If not, one of the key assumptions for Michaelis-Menten kinetics is invalidated and the authors should discuss this in the text.

It is unfortunate, but we cannot measure the true concentration of enzymes in the reactor. Based on the calculated units of enzyme immobilized on beads, only several units of enzymes (for most of the case less than 10 units) were used, and we can therefore safely assume that enzyme concentrations are well lower than the substrate concentrations and MM kinetics are valid. For example, in the case of enzyme PK, the specific activity of free PK we get from Sigma is 350-600 units/mg protein, 10 U of PK-beads in the flow reactor contains 1/35 – 1/60 mg proteins.

The authors claim HPLC error is the reason for poor predictions below 0.3 mM. It might be helpful to show a standard curve and perhaps a couple of representative traces at 0.3 mM and below to back this up. Perhaps more importantly, if there is indeed poor signal to noise below 0.3 mM, this should be apparent from multiple technical replicates for experiments 6 and 7? Do the authors have such data? Otherwise, it does seem more likely to me that the model fails at these low concentrations because the model structure is not quite right (as highlighted by myself and reviewer 2), but that this isn't important at higher concentrations (i.e. when operating above K_m).

As shown in Table S2, the std for experiments 6 and 7 are slightly higher than other experiments, but still at the same order (several tens of μM) with other experiments. However, as the total concentration is low, there is more noise than in other experiments, which makes the calculated prediction error high.

Table S2. Results of the final experiments

Exp	Output_1 (μM)			Output_2 (μM)			Output_Average (μM)					
	UTP	GTP	ATP	UTP	GTP	ATP	UTP	std	GTP	std	ATP	std
1	700	531	1573	769	580	1580	735	35	556	25	1577	4
2	342	257	1617	353	287	1502	348	6	272	15	1560	58
3	360	900	513	380	1063	489	370	10	982	82	501	12
4	250	364	395	337	332	422	294	44	348	16	409	14
5	219	287	125	265	212	85	242	23	250	38	105	20
6	206	57	55	197	148	149	202	5	103	46	102	47
7	295	74	284	116	53	149	206	90	64	11	217	68

Quite a lot of the data used to fit the model in Figure 3b appears to be below 0.3 mM, is this not an issue if HPLC signal to noise is a problem?

In so far that the model that we train works as presented, no. The general trends are consistent and the model maps onto these well, additionally for low concentrations the absolute difference in concentrations show clear trends i.e., the model is trained to recognize that the concentration is rising or dropping when we measure a range of time points in the low concentration regime. This was sufficient to train a predictive model. We sampled the regimes

in the final experiment as broadly as possible to simultaneously assess where the predictions start to break down, since the methods focus on experimental design to gain experimental control by training a useful model. The model is significantly more useful if you can show its limits (in this case we posit that these limits are mostly on the measurement side).

Reviewer #5 (Remarks to the Author):

In this manuscript, the authors have proposed a methodology based on active learning to identify optimal datasets to train kinetic models of enzymatic reaction networks. They applied this method to a network containing 8 reactions to modulate product ratios while maintaining the overall conversion efficiency.

Major comments:

Major comments:

1. Novelty: The focus of the manuscript appears to be on optimal experimental design, i.e., identifying the most informative datasets that maximize predictive capability. It is unclear what this method does differently than a simple k-fold cross validation that is routinely used to validate kinetic models of metabolism. Furthermore, kinetic models of reaction networks typically contain far more parameters than data available for parametrization. The authors must also clarify how their OED algorithm ensures statistical significance of the identified parameters (passing a χ^2 goodness-of-fit test) to avoid overfitting the training data.

We thank the reviewer for their question, to our understanding K-fold cross validation applies to an established dataset, whereby this data is split in different folds to test and train a model on these folds. You can subsequently quantify the degree to which certain parts of the data might or might not bias the model and or if the model is valid. This is a post-hoc analysis. The OED algorithm provides you with the most informative perturbations that need to be applied to your system first. Given a model that you define, the dataset can then be used to assess whether that model can pass a cross validation test (not directly, but indirectly by maximizing the information within the data about the kinetics).

To ensure the trained model is not biased/overfit over we embed the OED algorithm in the active learning cycle and test the model each iteration. The model trained on the data obtained in the previous iterations is tested by assessing the accuracy of its prediction of the experiment (fig 4). For each experiment we change the bead composition of the reactor, thereby altering the enzyme concentrations. The substrates concentrations are constantly perturbed, thus as a test dataset it challenges the model to be quantitatively accurate (the absolute error expressed as the percentage the prediction deviates from data is 10-15% for the metabolites measured, not orders of magnitude). For the final experiment, we show in figure S7 that the predictive power of the model improves every iteration i.e. from very bad to good.

2. Applicability to larger networks and other kinetic formalisms: The authors have applied their algorithm to a small model containing 8 reactions and 15 metabolites formulated using only the Michaelis-Menten type formulation. Several other formalisms are commonly used in the realm of kinetic modeling (Saa and Nielsen, 2017) that have a varying number of parameters. Small models such as the one presented in this manuscript are typically well behaved and do not contain the complex interactions that contribute to parameter identifiability issues that can adversely impact predictive capability. The authors must evaluate the compatibility of their method with other modeling formalisms and quantify the influence of model formalism on identified optimal datasets. The authors must also apply their method on larger metabolic models such as those available for *E. coli* and other toy networks (Dash et al., 2017; Gopalakrishnan et al., 2020; Khodayari et al., 2014).

We agree with the reviewer that applying the OED and active learning approach to larger networks would be very interesting, additionally, a deep dive into quantifying the informational content of the designed experiments based on different modelling formalisms could be relevant for future work. Unfortunately, it is beyond the scope of this paper, which sought to establish control over the bio-catalysis of an ERN in flow, to expand our *experiments* to large networks (many 10s of enzymes), as the purification of enzymes, optimising experimental conditions, finding suitable analysis methods and knowing which observables are available, etc., is a project on its own. However, in response to the next question, we do illustrate how our computational method is suitable for tackling much larger networks.

3. Computational scalability: The authors have not provided any details on the computational power and time required to identify the optimal training dataset. The authors must also evaluate the performance of their algorithm with larger networks. Examples of networks at various sizes are available from literature (Gopalakrishnan et al., 2020).

We investigated the speed that AMICI offers with respect to the size of the model and presented results in a new figure S1 (see below). To our knowledge AMICI and JAX are currently the best compiled ODE solver packages compatible with python and/or MATLAB (and C++). To our knowledge and based on remarks by the developers, AMICI is favoured as the fastest for very large models e.g. >1000 ODEs (and JAX smaller systems). We opted to select 2 models, our own and the E. coli core metabolism model, built new model building software, and modified the E. coli core network from Khodayari et al. 2014, and placed it in the context of our *in vitro* set-up, assuming all species can be controlled by an inflow syringe.

The speed of the simulation will be a function of the stiffness of the system and the size of the model. For the OED set-up we need to construct sensitivity equations, thus a model with 100 states and 100 parameters requires each agent in the optimization algorithm to solve 10.000 equations each iteration such that we can construct the sensitivity matrix and calculate the D-optimality (or other) criterion. For the purposes of this test we assume every species can be observed and all can be perturbed i.e. including the enzymes over a short timeframe (making the problem stiffer). For this we assume we want the sensitivity of all parameters to all states. In a real example we would likely solve an order of magnitude fewer equations since we cannot measure everything in practice (thus the sensitivities towards these states are meaningless and do not need to be calculated) and we do not need the sensitivities of all parameters (only those that need to be estimated).

We modified the Supporting Information to include a more comprehensive look at the computation times involved:

Figure S1. Computational speeds for different aspects of software presented in main text. a) Shows the difference between an ODE model solved in python and AMICI for the number of species the model simulates. b) Shows the time course data (12 hours) of a randomized pulse experiment for both the model presented in the main text (19 states simulated in this version) and the *E. coli* core model (119 states simulated). The input flow rates are altered every 15 minutes. c) Shows the convergence time for each model trained with either 6 agents or 30 agents. d) Breaks down c and shows the median CPU given the number of simulated species per iteration i.e. solving a model 6 times means implies solved 114 ODE's for the first model and 714 solved ODEs for the *E. coli* core model. e) Shows the relative convergence scores of these optimizations (log normalized summed least squares error), the inset shows the absolute convergence values of the worst fit. f) Shows the average simulation time of each model given the number of ODEs that need to be solved, for the OED version of the models i.e. including sensitivity equations this takes a lot longer. g) Shows the time it takes to optimize a flow experiment for each model utilizing a single core (100 iterations of the algorithm)..

The goal of the model module is to map an ODE model (written out in a human-interpretable format) to different open-source packages, specifically compilers that compile the model to a non-interpreted coding language like C++. This is highlighted by Figure S1 where a system of 13 ODEs solves 80 times faster in AMICI than python. To test the scalability of the entire active learning approach software we opted to test its speed applied to two models (the tests were run on a single Intel Xeon E5-1660 v4 @ 3.2 GHz core 2017); the network/model featured

in the main text and the *in vivo* metabolic core E. coli model from Khodayari et al. (2014), placed in an *in vitro* context.

Computational speed: models

We test the speed of both the training of the model and the optimal experimental design for 2 models:

- Main text model (free enzyme):
 - 20 reactions
 - 19 simulated species (all metabolites observed)
 - **1349** sensitivity ODEs
- E. Coli core model (free enzyme):
 - 100 reactions (modified and removed transport and degradation from model)
 - 119 simulated species (all metabolites observed)
 - **44982** sensitivity ODEs

We maintain a reactor set-up akin to the one used in the main text yet utilize 12 syringes (maximum without modification) that flow both enzymes and substrates into the reactor with different inflow rates every 15 minutes (making the problem stiffer). The summed inflow rate of these syringes determines the total outflow rate of the species within the reactor. For the purposes of this test we include all species, enzymes and metabolites (all observed) in the stock solution of the syringe, the inflow rates of individual syringes can change every 15 minutes. This means we do not allow the system to go towards steady state during the experiment, making the problem stiffer, thus increasing the computational load. For the E. coli model, the enzymes are divided over the first 10 syringes, the energy carrying molecules (Glucose, ATP, NADPH etc.), are combined in a single syringe, all other substrates and products in another, single syringe. For the original model each enzyme gets its own syringe.

Both models are initialized with random set of parameters and we build an *in silico* dataset by running a pulse experiment for 12 hours. We applied a series of randomized time dependent pulses to both networks (15 minutes intervals). This results in a simulated time course dataset for both models shown in Fig. S1b (left = main text model) and S1c (right = E. coli core model). The xml files for the model can be found at github.com/huckgroup/OED.

Computational speed: training a model

When we train the model we change a single hyper parameter for each, the number of agents that move across the fitness landscape (i.e., parameter mutations that are simulated each iteration). Figure. S1c shows the distribution of convergence times for each model. Summarized, a single CPU:

- ~80 seconds to converge (main text model trained by 6 agents).
- ~7.5 minutes to converge (main text model trained by 30 agents).
- ~25 minutes to converge (E. coli core model trained by 6 agents)
- ~ 155 minutes (E. coli core model trained by 30 agents)

In Figure S1d it breaks this down in median CPU time per the number of total ODEs per iteration that had to be solved for each model. In Figure S1e we show the relative convergence least squared error score for each model/optimization combination with the absolute error in the inset. This shows that a small number of agents leads to identical fits for a small model but, for the E. coli core model, more agents per iteration improve the fit (the translated *in vivo* model is significantly stiffer than the model from the main text).

Computational speed: optimal experimental design of control inputs

For optimal experimental design of the control inputs the sensitivity equations need to be solved to the sensitivity matrix. This equates to unique ODEs for every parameter and state combination. The model from the main text has 19 ODEs plus 1558 sensitivity ODEs, the E. coli core model has 119 ODEs plus 44982 sensitivity ODEs. In a practical example we will never have this many observables and we do not need the sensitivity of all parameters e.g. inflow rates to all states, which would drastically reduce the size of the sensitivity matrix. However, we assumed this was the case to test the OED software. We optimized the control inputs of 12 syringes for both models with 6 agents across 100 iterations. Figure S1f shows it takes less than 2 hours for the smaller model to converge on a single CPU whereas the larger model takes 58 hours. When we test the average simulation times of a single model this breaks down according to Figure S1e.

- Main text model: **0.078** seconds
- Main text model + sensitivity equations: **10.2 seconds**
- E. coli core model: **0.6** seconds
- E. coli core model + sensitivity equations: **339 seconds**

Summarized, these numbers give a rough indication of the computational times involved. We note that the current E. coli core model exceeds the scope of currently feasible *in vitro* applications yet still remains computationally solvable within the timeframe of a single experiment (even on a single core).

To achieve this transformation from *in vivo* to *in vitro*, we built a new model builder, which is included in the appendix as well.

We include model building software that can take a list of any enzymatic reaction network defined by the enzyme, its reversibility, the kinetic rate law, its substrates, its products, its inhibitors and activators. The software can be found on github.com/huckgroup/OED, see example of format below

'PK',	True,	'GH',	['ADP','PEP'],	['Pyruvate','ATP'],[],[]
'UMPK',	True,	'GH',	['UMP','ATP'],	['UDP','ADP'],[],[]
'GMPK',	True,	'GH',	['GMP','ATP'],	['GDP','ADP'],[],[]
'AK',	True,	'GH',	['AMP','ATP'],	['ADP','ADP'],[],[]
'APRT',	False,	'GH',	['PRPP','Adenine'],	['AMP'],[],[]
'UPRT',	False,	'GH',	['PRPP','Uracil'],	['UMP'],[],[]

We rewrote the methods section of the paper and added references of interest:

An overview for the software that performs the optimizations can be found in Supplementary Information 1. The package is written in Python 3.8 (python software foundation, Delaware US). A generated text-based model object25 is translated to an SBML and AMICI object modified from ref 28 and ref 42 (Supporting information fig S1-S4). AMICI is an ODE compilation package to C++ which is continuously updated.⁴³⁻⁴⁶ Code can be found at huckgroup github at <http://github.com/huckgroup/OED>. Several publically available tools integrate with AMICI⁴⁵⁻⁴⁹. This is needed for the expanding repertoire of ever larger kinetic models (most *in vivo*)⁵⁰⁻⁵⁴. To quantify the computational cost (and its general application to larger systems) we tested the speed of the pipeline presented on an *in vivo* metabolic core E. coli core metabolism model (ref 53) and placed it in the context of our *in vitro* reactor set-up (see Figure S1). This test was run on a single core of Intel Xeon E5-1660 v4 @ 3.2 GHz. For more information on the efficiency of AMICI itself (where the bulk of the calculations are performed) we refer the reader to^{43, 44, 55-57}, or its by now, numerous applications⁵⁸⁻⁶⁴.

We the following sentence to the main text paper.

(in supplementary figure S1 we probed the applicability of this software to larger systems, specifically by comparing CPU time needed for the model presented here and the E.coli core metabolism model).

4. Uncertainty analysis and quantification: Parameter identifiability and precision is a common issue with kinetic models due to their nonlinear and nonconvex structure. Although the authors have focused on product yield as their validating data, poor parameterization precision can adversely affect the predictive power, especially in models with perturbed enzyme levels. Figure 4a shows a poor resolution of Kcat and KM parameters for most reactions. This will impact the predictive power of networks in which enzyme levels are perturbed, which was not done in this study. The authors should include additional validation for experiments with perturbed enzyme levels to reinforce their analyses.

We agree with the reviewer that poor parameterization leads to adverse effects on predictive power. And we agree that the concentrations of the enzymes in the reactor need to be altered, hence why every iteration of the active learning cycle we change the bead composition of the reactor, thereby altering the enzyme concentrations. If we view the experiments as one continues 32 hour experiment, we effectively perturb all the enzyme concentrations 4 times, once every ~8 hours and again for the final experiment. Note the enzyme beads do not flow out of the reactor, only the substrates, thus pulse like inputs for the beads are not possible.

Perturbing the composition of enzymes in the reactor however is needed, in figure S10 we show that the model trained using the first experiment is heavily biased towards the training data and does not map the conversion

fluxes with respect to the enzyme concentrations at all (hence the prediction error that spans orders of magnitude when it attempts to predict a new experiment including a reactor with a different bead compositions).

The subsequent perturbations constrain the number of parameter combinations that can map onto the observed production fluxes for different reactor conditions. Because we test the models trained during each optimization cycle to predict the next experiment (i.e. this model is not trained for this enzyme concentration combination and has to deal with continuous time dependent inputs of multiple substrates as well), we know when the model has enough resolution to predict complex specific product ratios constrained by a specific conversion efficiency. So instead of focussing on identifying exact rates, this more utilitarian approach actively balances the identifiability of the rates and the complexity of the model, the predictive power of the model, and the available data. Iterating towards a model/dataset combinations can control the network (quantitatively) within the scope of our experimental set-up.

Reviewer #4 (Remarks to the Author):

The authors have addressed all my concerns.

Reviewer #5 (Remarks to the Author):

The authors have adequately addressed the previously raised concerns